# Structure-Based Design of Antivirals against Envelope Glycoprotein of Dengue Virus

**DOI:** 10.3390/v12040367

**Published:** 2020-03-26

**Authors:** Mohd Ishtiaq Anasir, Babu Ramanathan, Chit Laa Poh

**Affiliations:** 1Center for Virus and Vaccine Research, School of Science and Technology, Sunway University, Kuala Lumpur, Selangor 47500, Malaysia; ishtiaqa@sunway.edu.my; 2Department of Biological Sciences, School of Science and Technology, Sunway University, Kuala Lumpur, Selangor 47500, Malaysia; babur@sunway.edu.my

**Keywords:** dengue virus, structural biology, antiviral, envelope glycoprotein

## Abstract

Dengue virus (DENV) presents a significant threat to global public health with more than 500,000 hospitalizations and 25,000 deaths annually. Currently, there is no clinically approved antiviral drug to treat DENV infection. The envelope (E) glycoprotein of DENV is a promising target for drug discovery as the E protein is important for viral attachment and fusion. Understanding the structure and function of DENV E protein has led to the exploration of structure-based drug discovery of antiviral compounds and peptides against DENV infections. This review summarizes the structural information of the DENV E protein with regards to DENV attachment and fusion. The information enables the development of antiviral agents through structure-based approaches. In addition, this review compares the potency of antivirals targeting the E protein with the antivirals targeting DENV multifunctional enzymes, repurposed drugs and clinically approved antiviral drugs. None of the current DENV antiviral candidates possess potency similar to the approved antiviral drugs which indicates that more efforts and resources must be invested before an effective DENV drug materializes.

## 1. Introduction

Dengue virus (DENV) infection is the most prevalent arboviral disease with 390 million infections occurring annually, of which 96 million infections were manifested clinically [1]. Infection with any of the four DENV serotypes (DENV1-4) can produce a range of clinical manifestations, from a mild flu-like illness to life-threatening severe diseases known as dengue haemorrhagic fever (DHF) and dengue shock syndrome (DSS) in some individuals. Vaccines and antiviral therapies are being developed to prevent or treat DENV infection. The development of DENV vaccines has made significant progress with the licensing of CYD-TDV (Dengvaxia) in 20 countries including Philippines, Mexico, Brazil, El Salvador, Indonesia, Costa Rica, Paraguay, Guatemala, Peru, Thailand, and Singapore since 2015 [2,3]. However, Dengvaxia has failed to provide complete protection with serotype-specific efficacy of 51% for DENV1, 34% for DENV2, 75% for DENV3 and 77% for DENV4 [4]. Furthermore, the vaccine only exhibited good efficacy in seropositive individuals (60%–76% efficacy) and not in seronegative individuals (19%–39% efficacy) [5]. Dengvaxia was shown to increase the risk of dengue hospitalizations in seronegative vaccinees in comparison to unvaccinated seronegative individuals [6]. An increased risk of severe dengue has also been observed in seronegative vaccine recipients during subsequent infection with other heterologous DENV serotypes [5].

Similarly, the development of effective antiviral agents against DENV infection is being widely pursued. The discovery of antiviral agents against DENV was performed using multiple strategies such as the screening of natural compounds, small molecules, nucleoside analogues and chemical libraries. Additionally, there is a resurgence in the peptide-based drug research to develop effective antiviral peptides against viruses following the success of the commercially available enfuvirtide [7]. Enfuvirtide is a 36 amino acid peptide that mimics a region within the human immunodeficiency virus HIV-1 envelope glycoprotein 41 (gp41). Enfuvirtide blocks the fusion of HIV-1 and cellular membranes by binding to the HIV gp41.

## 2. DENV Genome and Encoded Proteins

DENV is a member of the *Flaviviridae* family. Its genome comprises a single-stranded positive-sense RNA of approximately 11 kb that encodes a polyprotein [8]. The polyprotein is cleaved by proteases into three structural proteins which are the nucleocapsid protein (C), envelope glycoprotein (E) and precursor membrane (prM) proteins, and seven non-structural proteins which are NS1, NS2A, NS2B, NS3, NS4A, NS4B and NS5 (Figure 1) [9]. The structural proteins form the virus particle with both prM and E being located at the surface while the C protein is located inside the envelope. The C protein is important for encapsidation that protects the DENV genome [10]. The prM maintains the spatial structure of the E protein [11]. The E protein has been implicated in the membrane fusion between host cells and DENV particles [11]. Importantly, non-structural proteins regulate various stages of the DENV lifecycle, including viral RNA replication, virion assembly, polyprotein cleavage, maturation, and defense against host immunity [12,13,14,15]. Antiviral development approaches thus far have targeted both structural and non-structural proteins, with the main focus on multifunctional enzymes such as NS3, NS4B and NS5 [16,17,18,19,20].

## 3. Current Status of Dengue Antiviral Development

Severe dengue including DHF and DSS are associated with a fatality rate as high as 15% in the absence of proper medical attention [21]. Supportive fluid therapy remains the only available treatment for severe dengue [22]. Therefore, there is an unmet need to develop a safe and effective DENV antiviral. The main rationale for the development of a DENV antiviral is to reduce viremia during the early phase of the infection, which is expected to prevent progression to DHF/DSS [23,24]. An ideal dengue antiviral must significantly reduce the viral load within 70 h of illness onset to prevent disease progression to DHF or DSS [24]. Furthermore, the drug should inhibit all four DENV serotypes as these serotypes could co-circulate in endemic areas [25,26].

### 3.1. Drug Repurposing for Dengue Therapy

A panel of drugs initially developed for other viral and non-viral diseases have been evaluated for their activities against DENV (Table 1). Drug repurposing strategy is favorable as it takes a shorter time and less cost to reach the clinic [27]. Many of the repurposed drugs have been evaluated for their efficacies to treat dengue patients. Four of the drugs (balapiravir, celgosivir, UV4B and ivermectin) exhibited sub-µM in vitro potency against DENVs, justifying their evaluations in the clinical trials (Table 1). Reductions in viremia and NS1 antigenaemia were used as the main endpoints in these clinical trials [20,28,29,30,31]. 

Chloroquine was evaluated against DENV due to its ability to inhibit endosomal acidification. This potentially interferes with the fusion of viral and endosomal membranes. In vitro studies showed that chloroquine inhibited DENV1 replication in THP cells overexpressing DC-SIGN and DENV2 replication in U937 and Vero cells [32,33,34]. Furthermore, chloroquine was shown to reduce viremia in monkeys infected with DENV2 [36]. Despite the promising pre-clinical findings, chloroquine did not reduce viremia and NS1 antigenaemia in dengue patients [28]. Chloroquine was not efficacious probably because it did not achieve sufficient inhibitory concentration within the reticuloendothelial cells where DENV replication occurred [28]. 

Corticosteroid is mainly adopted as an adjunct immunomodulatory treatment for diseases caused by immune responses [48]. Immune pathology of dengue has similarities to other autoimmune diseases that have been successfully treated by corticosteroids [48,62]. A clinical trial has shown the supportive evidence for the actions of methylprednisolone to prevent dengue progression to DHF or DSS [63]. Another clinical trial to evaluate the effects of immunomodulation by prednisolone revealed that the treatment with oral prednisolone during the acute stage of DENV infection did not lead to prolong viral clearance or other adverse effects [29]. However, the study also found that there was no reduction in the development of recognized complications of DENV infection [29]. The lack of clinical benefit could be due to limited immunomodulation achieved by prednisolone [37]. Similarly, statins were evaluated against DENV infection due to their anti-inflammatory and immunomodulatory effects. In particular, their stabilizing effect on vascular endothelium could be beneficial to prevent endothelial dysfunction in dengue patients [64]. Interestingly, in vitro studies revealed that lovastatin interfered with DENV virion assembly [38,40]. However, clinical trials revealed lovastatin did not reduce viremia in dengue patients [30]. Modipafant and ketotifen are other drugs that have been evaluated to treat dengue fever due to their anti-inflammatory properties. The clinical trial for modipafant (NCT02569827) is currently ongoing while the data for the clinical trial of ketotifen (NCT02673840) are yet to be available.

Balapiravir and Ribavirin are nucleoside analogues that were originally developed as antivirals against HCV infection. Balapiravir was found to be effective to reduce HCV viremia in dose-dependent and time-dependent manners [65]. However, further development of balapiravir for the treatment of chronic HCV infection has been stopped due to serious haematological adverse events observed in patients receiving prolong (2–3 months) treatment with balapiravir [66]. Researchers have evaluated the potential use of balapiravir as a treatment for dengue. This is due to the similarity in the architecture of its target protein, the RNA-dependent RNA polymerases between HCV and DENV. Although balapiravir demonstrated potent in vitro activity against DENV, no reduction in viremia and NS1 antigenaemia were observed in adult dengue patients [20]. The failure of balapiravir in the clinical trial could be due to the activation of peripheral blood mononuclear cells by DENV infection. This decreased the efficiency of the conversion of balapiravir to its active form, leading to decreased antiviral potency [47]. On the other hand, ribavirin displayed synergistic in vitro and in vivo anti-DENV2 effects in combination with an oxygenated alkyl imino sugar derivative, CM-10-18 [49]. Ribavirin was evaluated clinically in combination with traditional Chinese medicine in China with the data are as yet, unavailable (NCT01973855).

Iminosugars such as celgosivir and UV-4B have also been evaluated for their antiviral activities against DENV. Both compounds exhibited potent in vitro antiviral activities against DENV1-4 (Table 1) [52,53,56]. In addition, both drugs protected mice against DENV2 lethal challenge [55,57]. The first clinical trial of celgosivir revealed that there was no significant reduction of viremia or fever burden in dengue patients [31]. Another clinical trial to evaluate the efficacy of celgosivir with a revised dosing regimen is currently on-going and is expected to finish in September 2020 (NCT02569827) [54]. For UV-4B, the clinical trial has been terminated due to business reasons (NCT02696291). Lastly, the anti-parasitic drug, ivermectin, was identified to be a potent inhibitor of DENV replication [59]. However, preliminary results from a phase 2 clinical trial of ivermectin shown no reduction in viremia, despite exhibiting in vitro potency in the high nM range (500 nM). This suggested that compounds with higher in vitro potency (low nM to pM range) might provide better pharmaceutical properties.

### 3.2. Approved Drugs against Viruses in the Flaviviridae Family

There is no clinically approved drug against viruses within the Flaviviridae family except against the hepatitis C virus (HCV) [67]. Seven direct-acting antivirals have been approved by the Food and Drug Administration (FDA) for the treatment of chronic HCV infection [68]. Three combination regimens are effective against all HCV genotypes: (1) sofosbuvir and velpatasvir, (2) sofosbuvir, velpatasvir, and voxilaprevir and (3) glecaprevir and pibrentasvir [68]. Each of these drugs targets either NS5A, NS5B or NS3/4A. Sofosbuvir is an inhibitor of HCV NS5B polymerase. Its triphosphate exhibited potent inhibitory activity against NS5B with a K_i_ (inhibitory constant) of 0.42 µM [69]. Velpatasvir and pibrentasvir are pan-genotypic inhibitors of HCV NS5A with EC_50_ values in the range of 1 to 130 pM and 1.4 to 5.0 pM, respectively [70]. Voxilaprevir and glecaprevir are potent pan-genotypic HCV NS3/4A protease inhibitors with EC_50_ values in the range of 1–130 nM and 3.5–11.3 nM, respectively [71,72]. Importantly, the pre-clinical data demonstrated the high *in vitro* potency (nM to pM) required for antivirals to be successful. The potency in the low nM to pM range enables the drugs to be administered in lower dosages to achieve desired effects. Interestingly, sofosbuvir exhibited in vitro anti-DENV2 activity with an EC_50_ value of 1.4 µM [73]. Although the potency of sofosbuvir against DENV is not in the nM to pM range, it could be worthwhile to evaluate its potential as a repurposed drug to treat DENV infection in clinical settings. Another important lesson from the success of HCV therapies is that the combination of several drugs targeting different viral proteins could be mandatory to formulate effective anti-dengue therapies. 

### 3.3. Antivirals Targeting Dengue Proteins

Rational dengue antiviral discovery has been pursued to identify promising clinical candidates. The NS3/NS2B protease is one of the main target for dengue antiviral discovery with inhibitors such as compound 35 (IC_50_: 0.6 μM), 42a (IC_50_: 0.21 μM), 45a (IC_50_: 0.26 μM), SK-12 (IC_50_: 0.98 μM) and compound 18 (IC_50_: 0.38 μM) [74,75,76,77]. In addition, inhibitors of other targets such as NS3 helicase, RNA-dependent RNA polymerase NS5 and non-enzymatic NS4B with sub-μM potency have been discovered [78,79,80]. Besides, the C protein is being explored as a drug target. For instance, ST-148 inhibited DENV2 with potency in the low nM range (EC_50_: 30–50 nM) [81]. However, some key challenges could impede the entry of these compounds into clinical trials. Firstly, the identification of viral resistance towards these compounds could significantly hamper their further development. For instance, a single amino acid mutation at position 63 of DENV2 NS4B was found to confer viral resistance towards a highly potent NS4B inhibitor compound 1a (EC_50_: 0.012 µM) [80]. Secondly, some of these compounds lacked antiviral activities against the four DENV serotypes as is the case of compound 1a that inhibited DENV2 and 3 only [80]. Thirdly, the development of compounds that have safety issues will be terminated. For example, an adenosine nucleoside inhibitor NITD-008 exhibited potent antiviral activities against DENV2 with an IC_50_ value of 0.31 μM [82]. However, animal studies using rats and dogs revealed that two weeks of NITD-008 oral dosing led to severe side effects in both animals such as weight loss, blood abnormalities and movement disorders [82]. Fourthly, some compounds lack potent anti-DENV activity when evaluated in a murine model. For example, ST-610 that potently inhibited NS3 helicase (EC_50_: 0.272 μM) exhibited a modest reduction of viremia (5.2 fold) in DENV2-infected mice, signifying further optimization is needed to improve its efficacy in vivo [78]. Lastly, compounds with poor pharmacokinetic properties are not suitable to enter clinical trials without further optimization. NITD-618 is a potent NS4B inhibitor that was found to be active against the four DENV serotypes with EC_50_ values ranging from 1 to 4.1 μM [83]. Unfortunately, NITD-618 has poor pharmacokinetic properties which prevented the evaluation of its activity in a murine model [7]. Similarly, the development of ST-148 was stopped due to limited oral bioavailability and rapid clearance [7]. In addition, the inability of ST-610 to significantly reduce viremia in DENV2-infected mice could be due to its poor pharmacokinetic profiles in the murine model. Although these challenges have halted the development of some of these compounds, many other compounds are yet to be developed and evaluated. 

## 4. Discovery of Antiviral Agents Targeting Dengue Envelope Glycoprotein

The inhibition of attachment and fusion is another feasible strategy to neutralize DENV infection. The E glycoprotein is the main mediator of DENV attachment and fusion. It belongs to a class II viral membrane fusion protein [84]. The E protein adopts two main conformations which are 1) dimeric in mature DENV and 2) trimeric in the immature DENV and in the acidic environment of the endosome during fusion (Figure 2). In the mature virion, the antiparallel dimeric E proteins lay flat on the virion surface [85,86]. In the immature DENV and fusion intermediates, the E proteins form trimers that are pointing away from the DENV membrane, giving the virion a spiky appearance. The E protein is composed of three ‘discontinuous’ distinct domains in the N-terminal region (Figure 2A). The central domain 1 or EDI is formed by eight β-strand barrels. The domain 2 or EDII is an elongated domain containing a fusion loop that is conserved among all flaviviruses. Three hydrophobic amino acids (W101, L107 and F108) within the fusion loop are essential for the insertion of trimeric E protein into the endosomal membrane during fusion [87]. These residues are exposed at the tip of the trimeric E protein [88]. In contrast, these residues are buried at the dimer interface in the mature virion [85]. The domain 3 or EDIII is an immunoglobulin-like domain that is responsible for the cellular receptor binding [89]. EDIII connects the three N-terminal domains to the stem region. The stem region is part of the outer lipid leaflet of the viral membrane. It is composed of two α-helices connected by a conserved sequence [90]. The position of these helices rearranges during fusion. Following the stem region is the C-terminal transmembrane domain. This domain comprised a pair of antiparallel coiled-coils TM1 and TM2, which contribute to the stability of the trimeric E protein and the completion of flaviviral fusion with the endosomal membrane [91].

The mechanism of DENV-cellular membrane fusion has been proposed. Firstly, DENV attaches to the cells via the interaction of its E protein with cell-surface receptors (Figure 3). Several studies revealed that the region spanning amino acids 305 to 315 (in particular K305, K307, K310) and 382 to 385 within EDIII carried the determinants for cellular receptor binding [94,95,96,97,98,99]. In addition, the E protein was found to interact with β3 integrin through two regions within EDIII, spanning amino acids 333–347 and 381–394 [100]. A complex structure of DENV binding to dendritic cell-specific intercellular adhesion molecule-grabbing non-integrin (DC-SIGN) showed that the interaction of the complex occurred solely through the N67 glycan of DENV and carbohydrate recognition domain of DC-SIGN [101]. Multiple other amino acids, in particular K291 within EDI and K295 within the EDI-EDIII hinge (based on DENV2 sequence), have been demonstrated to mediate interactions with cellular receptors such as heparan sulfate (Figure 4) [94,102].

The attached DENV will be internalized into an endosome via the clathrin-dependent process. The reduced pH in the endosome causes E dimers on the virion surface to dissociate [85,88,93]. Subsequently, the EDII of the dissociated E protein hinges outwards from the viral surface, allowing the interaction of the fusion loops with the target endosomal membrane [88,93,103]. The interaction of the fusion loop with the host endosomal membrane promotes the formation of E protein trimers [104]. The trimerization process is mediated by significant domain rearrangements. During trimerization, the EDIII rotates by about 70° towards the endosomal membrane and interacts with the interfaces formed by EDI and EDII [88]. This movement brings the EDIII closer to the fusion loop. The trimers bridge the endosomal and viral membranes (Figure 3). The driving force of the pinching of the two membranes is mediated by the EDIII folding back to EDI and by the contacts between EDII and the stem region (Figure 5) [93,104]. The refolding of the E protein creates bending and hemifusion of both membranes, ultimately leading to the formation of lipidic fusion pores [105]. The pores allow the escape of the viral nucleocapsid into the host cytosol where the expression of viral RNA genome took place. An in-depth understanding of the attachment, entry, and fusion is paramount to design antivirals against DENV infection.

## 5. Structure-Based Discovery of Antiviral Compounds Targeting the E Protein

The main advantage of targeting the E protein is that drugs can engage the E protein extracellularly and eliminating the need to design membrane permeable drugs [106]. In addition, non-membrane permeable drugs that target the E protein extracellularly might display low toxicity as they did not act on intracellular proteins [107]. This is a clear advantage over drugs targeting viral enzymes such as nucleoside analogues that were prone to target intracellular proteins and caused unwanted off-target effects [108]. Many dengue researchers adopted the structure-based antiviral approach to discover antiviral agents targeting the envelope glycoprotein. This approach can be divided into two categories which are the structure-based virtual screening and the de novo ligand design (Figure 6). Structure-based ligand optimization is often incorporated into these approaches to increase the efficacy of the ligands. The antiviral candidates will be optimized based on the structure of their binding pockets. The initial candidates will serve as templates from which functional groups will be added to the candidate compounds to improve affinity and specificity to the target binding site.

### 5.1. Structure-Based Virtual Screening for Small Molecule Compounds Targeting the Hydrophobic Pocket in the E Protein

The existing structural and functional information of DENV E protein has facilitated the screening for DENV antivirals [109,110,111,112,113]. In particular, the crystal structure of the E protein ligand-binding pocket occupied by a detergent molecule n-octyl-β-D-glucoside (β-OG) has inspired many researchers to screen for molecules that could target this pocket (Figure 7) [85]. The β-OG binding altered the conformation of the kl hairpin, which shifted towards the dimer interface, forming a salt bridge and a hydrogen bond with the amino acids of its dimer partner [85]. The hydrophobic pocket was proposed to act as a hinge point in the fusion-activating conformational change upon pH reduction. Compounds blocking the hydrophobic pocket were postulated to interfere with conformational changes in the E protein during DENV–host membrane fusion [109,110,111,112,113].

Importantly, the study to evaluate DENV2 resistance towards compounds targeting the hydrophobic pocket revealed that the natural barrier to resistance might be high for these compounds [106]. Only one mutation (M196V) within the hydrophobic pocket was identified after more than ten passages with eight individual compounds [106]. This indicates that this pocket is suitable to be a drug target as it has a low mutation rate. Sequence analysis revealed that the hydrophobic pocket is highly conserved in all four DENV serotypes [85]. The four DENV serotypes possess identical hydrophobic amino acids V130, L135, F193, L198, and F279 that form the hydrophobic pocket. Thus, drugs targeting the hydrophobic pocket might be able to neutralize all four DENV serotypes. 

Researchers have performed molecular docking to screen databases containing more than 100,000 compounds for binding towards the β-OG pocket [109,110,111,112,113]. Subsequently, they performed various antiviral activity and fusion inhibition assays to screen for compounds with anti-DENV activity in vitro [109,110,111,112,113]. Biophysical techniques such as nuclear magnetic resonance (NMR) were utilized to measure the interactions of the compounds with the dimeric or trimeric E protein [110].

The earliest study using this approach was performed by Yang et. al. (2007), whereby molecular docking was utilized to screen for medicinal compounds against the β-OG pocket followed by DENV inhibition assays [109]. Two tetracycline derivatives, rolitetracycline and doxycycline, were identified to exert modest inhibition of DENV propagation with IC_50_ of 67.1 µM and 55.6 µM, respectively. Subsequently, compound P02 was identified from structure-based screening against DENV E protein [110]. The antiviral activities of P02 were measured based on its inhibition of YFV growth (IC_50_ of 13 µM) and replication (IC_50_ of 17 µM) indicating that PO2 might target the E protein and the non-structural proteins involved in genome replication or protein synthesis. A binding study using NMR revealed that P02 bound specifically to the β-OG pocket of purified DENV E protein. 

Other compounds that have been identified from virtual screening included NITD448, A5 and R1 [111,112,113]. These compounds were evaluated for their antiviral activities against DENV2 with reported IC_50_ values of 6.8 µM, 1.2 µM and 4 µM, respectively. Furthermore, A5 was also evaluated against other viruses including WNV, YFV and RSV [112]. A5 exerted potent antiviral activities against WNV and YFV with IC_50_ of 3.8 µM and 1.6 µM, respectively. Some of the anti-DENV compounds have been demonstrated to hinder DENV-host membrane fusion. For instance, A5 inhibited the low pH induction of syncytia (quantified as a fusion index) in infected insect cells expressing DENV E protein on the cell surface [112]. In addition, NITD448 was shown to inhibit membrane fusion between DENV2 and liposomes [111]. 

To understand the molecular basis of fusion inhibition by anti-DENV compounds, Wispelaere et al. (2018) mapped the binding sites of several small molecule compounds, viz. 2,4-disubstituted pyrimidine, 4,6-disubstituted pyrimidine and cyanohydrazone [106]. Using a photo-crosslinking approach, they mapped residues 247–284 of the hydrophobic pocket to be the binding site of the fusion inhibitors. Furthermore, mutation of a residue (M272S) within the pocket was found to abolish the binding of the inhibitors. These data supported the model that the fusion inhibitors bind to the hydrophobic pocket to exert their antiviral effects.

Structure-based optimization has been utilized to improve the antiviral activity of compounds identified from virtual screening [114]. For instance, Wang et. al. (2009) performed molecular docking coupled with antiviral assays to screen for compounds that could inhibit DENV2 in human cell lines [114]. They identified compounds 1 and 2 to inhibit DENV replication with EC_50_ of 1.69 µM and 0.90 µM, respectively. Thereafter, they identified functional groups that could be substituted to improve potency through structure-activity analysis. The synthetic chemistry efforts yielded compound 6 which displayed enhanced potency. Compound 6 inhibited all four DENV serotypes with EC_50_ values ranging from 0.068 µM to 0.496 µM. Furthermore, compound 6 also inhibited other flaviviruses such as YFV, WNV and Japanese encephalitis virus (JEV) with EC_50_ of 0.47 µM, 1.42 µM and 0.564 µM, respectively.

### 5.2. De Novo Ligand Design Targeting the DENV E Protein

De novo ligand design approaches have also been utilized to identify new compounds targeting the hydrophobic pocket of the E protein. Recently, Leal et. al. (2019) constructed a compound library comprising novel compounds postulated to have the ability to bind to the hydrophobic pocket similar to the β-OG molecule using the ligand-growing program Biochemical and Organic Model Builder (BOMB) [115,116]. They identified the 2,4-pyrimidine scaffold as the best candidate to be further developed as one of the molecules displayed antiviral activity in the µM range. Further optimization using a molecular dynamic-based approach ultimately yielded two compounds 3e and 3h, with high potency and good physicochemical properties. Both compounds were active against DENV1-4 with EC_50_ values in the range of 0.39–2.5 µM.

In general, the structure-based anti-DENV design approach has yielded arrays of compounds exhibiting low potencies against DENV and other flaviviruses such as YFV, WNV and JEV (Table 2). Only compounds 6, 3e, and 3h demonstrated modest in vitro potencies in the high nM range which are comparable to the approved drugs against HCV infection such as sofosbuvir (K_i_: 0.42 µM) [69]. Therefore, screening and design of more compounds are required to discover compounds with higher potency in the low nM to pM range. Furthermore, the atomic structure of antiviral compounds in complex with the hydrophobic pocket is yet to be determined. Atomic structures of the complexes are imperative to optimize the affinity and specificity of the compounds towards the hydrophobic pocket which may enhance the overall potency of promising compounds. It is also important to note that the potency of compounds targeting the E protein is lower than the overall potency of compounds targeting DENV multi-functional enzymes such as NS3 and NS5 proteins. However, the discovery of antivirals targeting the E protein should not be abandoned as a combination of drugs targeting multiple DENV enzymes and structural proteins might be required to formulate an effective regimen to treat DENV infection. 

There are several key challenges associated with the structure-based antiviral compound discovery. Firstly, this approach is mainly effective to identify compounds with promising antiviral activities, while the absorption, distribution, metabolism, excretion and toxicity (ADMET) prediction remains a problem [117]. Nonetheless, various computational methods are currently being incorporated in the structure-based drug design to improve ADMET prediction [118,119]. Secondly, the flexibility of target molecules remains a hurdle for molecular docking. Several approaches have been proposed to minimize this problem such as soft docking and induced-fit docking [120,121,122]. However, these approaches could not simulate large conformational changes efficiently. Therefore, other techniques such as molecular dynamics simulation should be incorporated into virtual screening to improve the efficiency of the screening. The incorporation of molecular dynamics was proven to be useful as the study incorporating molecular dynamics identified two of the most potent anti-DENV compounds 3e and 3h [116].

## 6. Structure-Based Discovery of Anti-DENV Peptides Targeting the E Protein

The use of peptides as antiviral agents is an attractive alternative especially with the success of FDA-approved enfuvirtide. There are also other initiatives such as the Virus Pathogen Resource (ViPR) that support the sequence-based predictions of antiviral peptides. Despite the number of peptides with potent antiviral activity being low, further research in this field is warranted as there is a possibility of discovering highly potent antiviral peptides similar to enfuvirtide with potency in the pM range. Additionally, antiviral peptide research will increase the repertoire of drugs targeting the structural and non-structural proteins of DENV. This will contribute to the drug combination approach to treat DENV infection. 

### 6.1. De Novo Design of Antiviral Peptides against DENV Infection

The de novo design approach has been used to develop peptides that can inhibit DENV. For instance, several short peptides targeting the hydrophobic pocket of the E protein were designed using this approach [123]. A two amino acid glutamic acid-phenylalanine (EF) peptide displayed the highest in vitro antiviral activity against DENV2 with an IC_50_ of 96.50 µM [123]. However, it exhibited weak inhibition against the other three DENV serotypes. In another study, BioMoDroid was utilized to design antiviral peptides to target the EDIII of DENV2 [124,125]. Two peptides, DET2 and DET4, exhibited inhibition of DENV2 with IC_50_ of 500 µM and 35 µM, respectively. Visualization using transmission electron microscopy showed that the surface of the virus particles treated with either of the peptides became uneven with rough edges distinct to the smooth outer surface of the untreated viral control. The disruption of the envelope structure potentially impeded viral attachment and entry. Several peptides were designed based on the non-native sequences derived from the E protein regions that are highly stable with regards to the structure and binding as evaluated by an all-atom scoring function (RAPDF) [126]. RAPDF evaluated the substitution of each amino acid in the selected region with each of the 19 naturally occurring amino acids. This approach identified four E protein regions with the potential for the highest stability and in situ binding affinities. The 1OAN1 peptide, which corresponded to the amino acids 41–60 of the E protein exhibited the most effective DENV2 inhibition with an IC_50_ of 7 µM and maximum inhibition of 99% at the concentration of 50 µM [126].

### 6.2. Rational Design of Anti-DENV Peptides Based on the Structure of the E Protein

In the structure-based rational design approach, researchers designed new peptides mimicking the specific regions in DENV such as the envelope glycoprotein, capsid protein and viral enzymes based on the available 3D structures. The E glycoprotein can be used as a template to design anti-DENV peptides considering our extensive knowledge of its function during DENV attachment and fusion. There are arrays of peptides derived from several regions of the E protein including the stem region and EDIII [100,126,127,128,129]. These peptides have been found to either directly interact with the E protein to disrupt the viral particle and fusion or by occupying the host cell surface receptors.

Based on the knowledge that the stem region is folding back towards EDII during membrane fusion, stem-derived peptides have been evaluated for their antiviral activities. A peptide derived from the sequence of the stem region named DN59, inhibited more than 99% of infection by DENV2 and WNV at a concentration of less than 25 µM [127]. The ability of DN59 to cross-inhibit WNV highlighted another advantage of using peptides as antivirals. Peptides that are derived from highly conserved regions across several viruses could potentially inhibit other viruses in the same family as well. In addition, the peptide exhibited broad antiviral activities against all four DENV serotypes [128]. The direct interactions of DN59 with the viral particles formed holes in the envelope and led to the release of the RNA genome. Intriguingly, the mechanism of action of DN59 is not as expected because stem-derived peptide should only bind to the virion after the dimeric E protein rearranged to form trimers in the low pH environment of an endosome. Nonetheless, there is a possibility that the peptide was able to access its binding site at the dimeric prefusion virions through the dynamic breathing of the virus particle at 37 °C [130,131,132]. 

Another group also designed antiviral peptides based on the stem region of DENV E protein [129]. In contrast to DN59, these peptides were shown to bind specifically to the trimeric E protein and blocked viral fusion. A two-step inhibition mechanism was hypothesized [129,133]. Firstly, the peptides form reversible non-specific contacts with the viral membrane, bringing the virion-bound peptides into the endosome. Secondly, the peptides within the endosome bound specifically and tightly to the binding site at the conformational intermediate of the E protein during the fusion, ultimately blocking membrane fusion.

In addition, researchers have designed antiviral peptides to block the interaction of viral particles with cellular receptors. In a study to understand the interactions of EDIII with integrin β3, two peptides, P4 and P7, were designed based on the amino acids of EDIII at the FG loop [100]. The two peptides exhibited antiviral activities against DENV2 in a dose-dependent manner with IC_50_ of 19 µM and 13 µM, respectively. Binding studies revealed that both peptides were able to bind to integrin β3. Subsequent entry inhibition assays revealed that P7 could inhibit the entry of both DENV1 and DENV2. In contrast, P4 only inhibited the entry of DENV2 but not DENV1. This highlighted the complexity of designing a peptide to inhibit all four DENV serotypes.

### 6.3. Structure-Based Optimization of Antiviral Peptide Candidates

The peptides that showed minimal antiviral activities could be further optimized using the de novo approach. In this approach, computational optimization using programs such as RAPDF was utilized to identify possible amino acids within the peptide that could be substituted to improve binding to their targets [134]. For instance, two peptides corresponding to the overlapping amino acids 205–223 (DN57) and amino acids 205–232 (DN81) within the EDII hinge region did not show substantial DENV inhibition [127]. In contrast, analogous West Nile virus EDII hinge region peptide potently inhibited WNV infection [127]. After computational optimization, DN57opt and DN81opt were scored by RAPDF to have improved structural stability and in situ binding when compared to the wild type sequences [126,134]. Both peptides displayed improved DENV inhibitory activities than the wild-type, with DN57opt and DN81opt exhibiting IC_50_ values of 8 µM (14-fold increase) and 36 µM (2-fold increase), respectively. Cryo-EM revealed that DN57opt changed the surface of the virions. This indicated that the mechanism of inhibition is most likely through the displacement of the EDII hinge region, leading to the trapping of the viral E protein in certain conformations that were unfavorable for viral binding and entry.

In general, the anti-DENV peptides displayed lower potencies (>2 µM) than the small molecule compounds targeting DENV E protein and enzymes (sub µM range) (Table 3). Furthermore, the potencies of these peptides were significantly lower than the potency of enfuvirtide, which inhibited HIV-1 infection with an IC_50_ value of 100 pM [135]. Nonetheless, modifications of the peptides such as the addition of a cholesterol moiety could improve their potency. This strategy was successful during the development of C34 against HIV and a 36-mer peptide against human parainfluenza, Hendra virus and Nipah virus [136,137]. Cholesterol tagging improved the potency of these peptides by enhancing the local concentration of the peptides at the membrane fusion site and the half-life of a peptide through its interactions with serum proteins [138]. Additionally, it was observed that chemical compounds targeting the E protein displayed a broader spectrum of activity against the four DENV serotypes than the antiviral peptides. This is likely due to the larger molecular size of peptides resulting in greater specificity for their binding sites. Thus, the design of DENV antiviral peptides should consider the amino acid sequence conservation among the four DENV serotypes to enhance their antiviral spectra. Despite their lower potencies and narrow spectra, the development of antiviral peptides against the four DENV serotypes is warranted. They have several advantages over chemical compounds as they have low toxicity, easy to synthesize and have low off-target effects due to their specificity and selectivity. Additionally, the molecular target on the E protein such as the binding site of the stem region is too large to be accommodated by chemical compounds. Therefore, high molecular weight biological molecules such as peptides are required to target the large protein-protein interaction sites. Besides that, chemical compounds are highly susceptible to viral resistance through mutations at the binding site. In contrast, peptides exhibited a higher barrier to resistance which is likely due to their extensive interactions with multiple amino acids on the target binding site [139].

## 7. Conclusions

Dengue is one of the most important infectious diseases causing significant public health threats. Therapeutic drug development will remain important since there is no clinically approved antiviral drug and the only licensed vaccine, Dengvaxia, is shrouded in controversy. Although no antiviral agent has been demonstrated to be effective against acute dengue in clinical trials, the pre-clinical research pipeline contains many antiviral agents that are promising. The availability of the DENV E protein structure has provided a foundation for structure-based drug discovery to identify antivirals against DENV. However, many of the antiviral agents have not been further developed. This is due to their low potency, poor pharmacokinetic properties and their inability to inhibit all four DENV serotypes. Therefore, future studies should prioritize the in vitro and in vivo evaluations of antiviral candidates against all four DENV serotypes. The ADMET and viral resistance of these candidates need to be established before their efficacies can be evaluated in clinical trials. More importantly, most of the compounds and peptides require further optimization to increase their potencies to be comparable to the potency of other clinically approved antiviral drugs. Although most of the current antiviral candidates will not enter clinical trials due to their low to modest potencies, these antivirals have provided a platform for future discovery of highly potent anti-DENV agents.

## Figures and Tables

**Figure 1 viruses-12-00367-f001:**
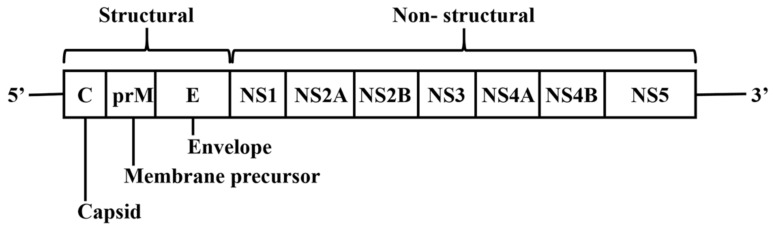
Schematic representation of the DENV genome and the encoded proteins.

**Figure 2 viruses-12-00367-f002:**
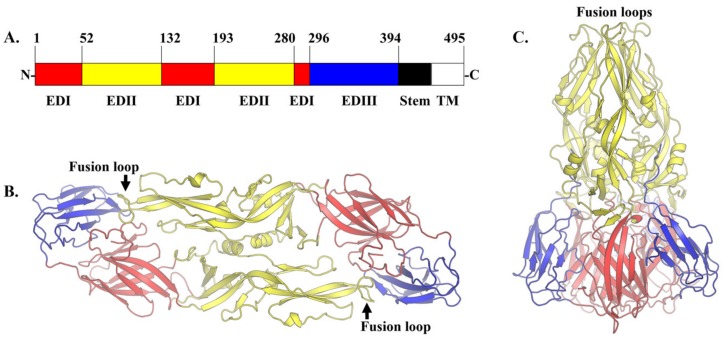
The envelope (E) glycoprotein. (**A**) The discontinuous domains of dengue E protein. Domain I (EDI) is in red, domain II (EDII) is in yellow and domain III (EDIII) is in blue. The stem region and the transmembrane (TM) domain spanning amino acids 394–495 are represented in black and white, respectively. (**B**) The E protein dimer in the mature DENV2 virion and in an environment with neutral pH (PDB ID: 1OAM). (**C**) The structure of trimeric DENV1 E protein in its postfusion conformation. The domains have rotated and shifted with EDIII undergoes the most significant rearrangement with 70° folding towards EDI and EDII (PDB ID: 4GSX). The fusion loop is buried at the dimer interface in mature virion. In contrast, the fusion loop is exposed at the tip of the E protein trimer in immature virion and during fusion.

**Figure 3 viruses-12-00367-f003:**
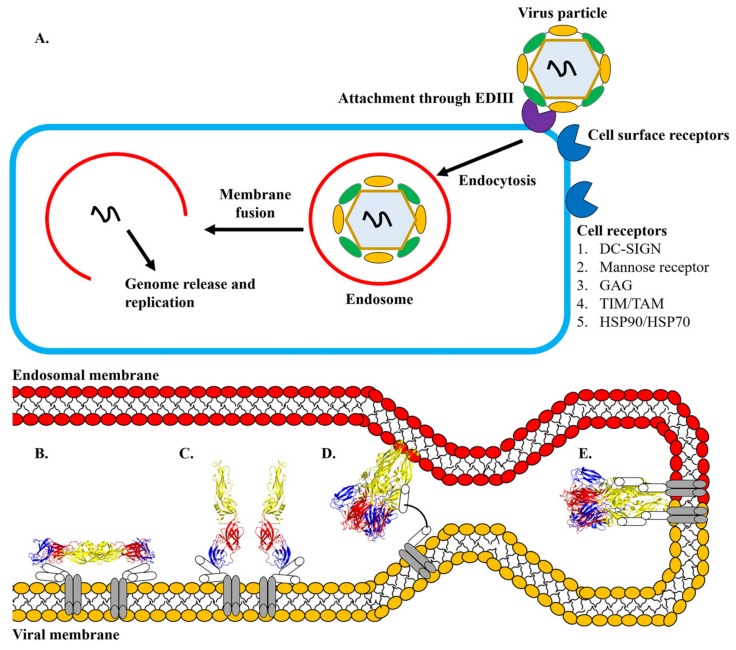
DENV attachment and fusion (PDB IDs: 1OAM and 4GSX). (**A**) Endocytic entry pathway of DENV. B-E) Proposed mechanism for DENV-host cell membrane fusion. The stem region of E protein is in white while the transmembrane anchor is in grey. The EDI, EDII and EDIII are in red, yellow and blue, respectively. (**B**) The binding of E protein to the receptor leads to the internalization of the DENV into an endosome. Reduced pH in the endosome causes the rearrangement of the E protein from dimer to trimer. (**C**) During the rearrangement, the tip of the EDII hinges outwards from the virion surface, allowing the fusion loop to be inserted into the cellular membrane to bridge cellular and viral membranes. (**D**) The bending and hemifusion of the membranes are mediated by the contacts made between EDIII-EDI and EDII-stem region. (**E**) Finally, the fusion loop makes contact with the transmembrane domain, completing membrane fusion and formation of pores. Figure 3A was modified from Hidari *et. al.* (2013) and Figure 3B-E was modified from Klein *et. al.* (2013) [92,93].

**Figure 4 viruses-12-00367-f004:**
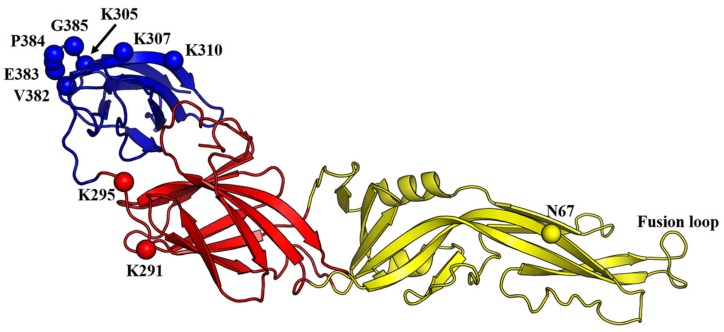
Locations of the amino acids postulated to be important for DENV2 attachment to the cells. The amino acids are shown as spheres in light grey and labelled. The EDI, EDII and EDIII are shown in red, yellow and blue, respectively (PDB ID: 1OAM). The majority of the amino acids that were implicated in DENV attachment to the cellular receptors are located in the EDIII.

**Figure 5 viruses-12-00367-f005:**
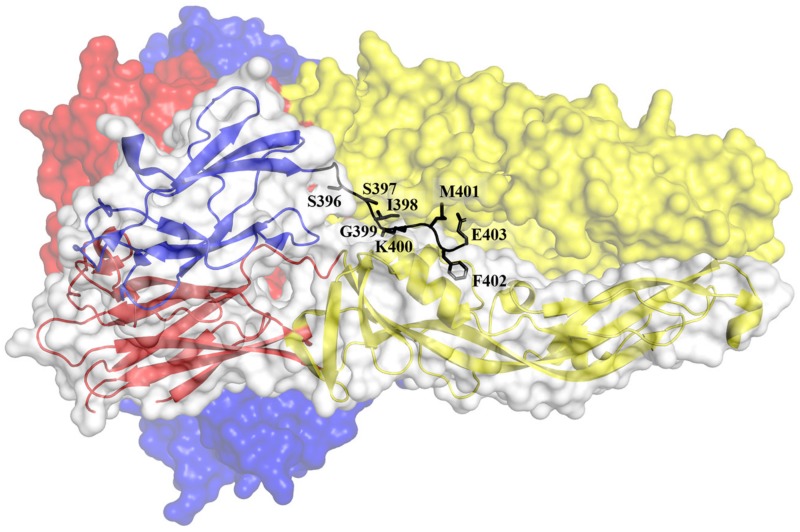
Stem region interactions with the groove formed by adjacent EDIIs in the trimeric DENV1 E protein (PDB ID: 4GSX). EDI, EDII and EDIII are represented in red, yellow and blue, respectively. The amino acids of the stem region are represented as sticks and labelled. The key hydrophobic amino acids of the stem region including I398, M401 and F402 interacted with the hydrophobic pocket formed by L216, P217, L218 and M260 of the EDII [93]. The figure was modified from Klein *et. al.* (2013) [93].

**Figure 6 viruses-12-00367-f006:**
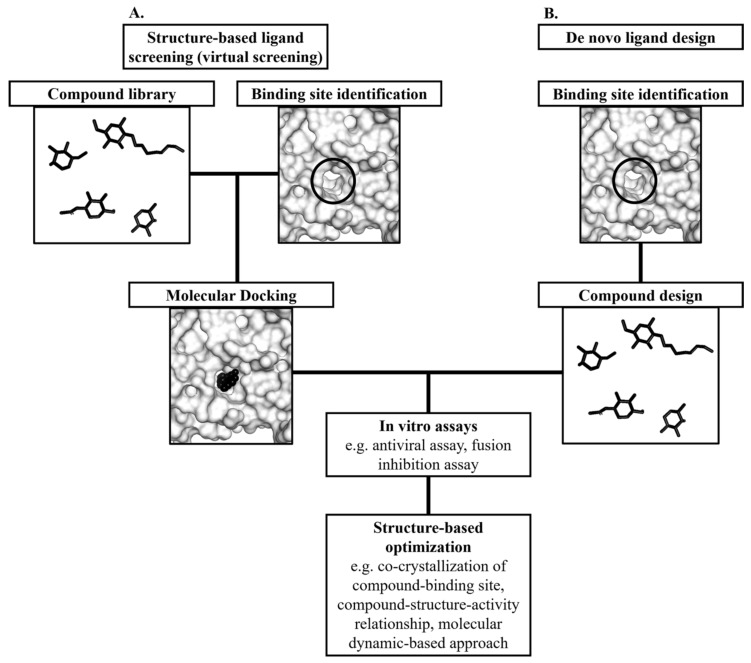
Structure-based dengue antiviral design. There are two approaches in structure-based antiviral design which are the structure-based ligand screening and the de novo ligand design. (**A**) In the structure-based ligand screening (virtual screening), molecular docking will be performed to identify compounds that can bind to the target binding site (circled). Thereafter, the compound hits will be assessed through in vitro assays such as antiviral assay. (**B**) In the *de novo* ligand design, compounds will be designed computationally to fit the target binding site (circled). Thereafter, in vitro assays will be performed to evaluate the antiviral activities of the compounds. Both approaches often require structure-based optimization to improve druggability and efficacy.

**Figure 7 viruses-12-00367-f007:**
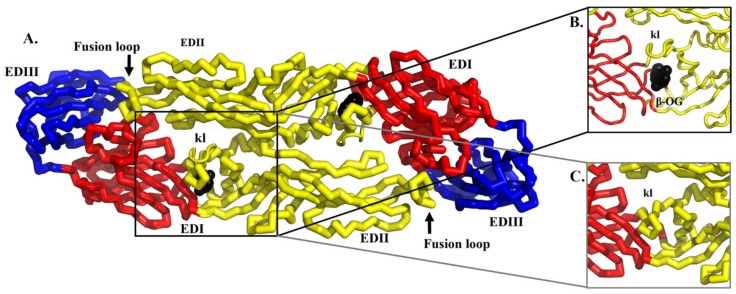
The superimposition of the structures of DENV2 E protein in the absence (ribbon) and presence (tube) of a detergent molecule. (**A**–**C**) The detergent molecules β-OG are shown in black. The domains of the E protein were labelled as EDI, EDII and EDIII, and colored as red, yellow and blue, respectively. The fusion loop is indicated by an arrow and labelled. The kl hairpin is labelled. The key difference between the two structures is the position of the kl hairpin. (**B**) In the presence of detergent, the kl hairpin will open up the hydrophobic pocket under the kl hairpin and the detergent can bind to the hydrophobic pocket (PDB ID: 1OAM). (**C**) In the absence of detergent, the kl hairpin is in the “closed position” and the hydrophobic amino acids are buried in the pocket (PDB ID: 1OAN). The figure was modified from Modis et. al. (2003) [85].

**Table 1 viruses-12-00367-t001:** Repurposed drugs evaluated in clinical trials.

Compound	The Rationale for Drug Repurposing	In Vitro Potency	Clinical Trial Status and Results	Ref
Chloroquine	Inhibited DENV entry and replication in vitro and in vivo.	N/A	No reduction in viremia and NS1 antigenaemia (ISRCTN38002730). Reduced pain intensity in patients (NCT00849602).	[28,32,33,34,35,36]
Prednisolone	Corticosteroid therapy could prevent the development of serious dengue complications.	N/A	No reduction in the development of complications of dengue (ISRCTN39575233).	[29,37]
Lovastatin	Exhibited modest inhibition of DENV2 replication in A549 cells. Protected mice from DENV2 infection.	EC_50_: 6.20 μM	No reduction in viremia (ISRCTN03147572). Minimal effect on dengue severity.	[30,38,39,40,41,42,43]
Modipafant	Inhibitor of platelet-activating factor receptor. Protective against lethal DENV2 challenge in mice.	N/A	The clinical trial is ongoing (NCT02569827)	[44]
Ketotifen	Exhibited inhibition of the DENV3-induced cell degranulation. Limits vascular leakage in mouse models of DENV2 infection.	N/A	Data not yet available (NCT02673840).	[45,46]
Balapiravir	Inhibited DENV4 RNA synthesis in vitro.	IC_50_: 0.89 μM	No reduction in viremia levels and NS1 antigenaemia (NCT01096576).	[20,47]
Ribavirin	Demonstrated synergistic effects in combination with an imino sugar CM-10-18 to suppress DENV2 replication in vitro and in the murine model.	EC_50_: 3 μM	Evaluated in combination with Chinese herbal drug (NCT01973855). Data are not yet available.	[48,49,50,51]
Celgosivir	Exhibited sub-μM in vitro anti-DENV1-4 activity and protective against DENV2 lethal challenge in mice.	EC_50_: 0.22- 0.68 μM	No significant reduction of viremia or fever burden (NCT01619969). Phase 2a clinical trial with an altered regimen of celgosivir is ongoing (NCT02569827).	[31,52,53,54,55]
UV-4B	Exhibited inhibition of DENV1-4 in vitro. Protective against lethal DENV2 challenge in mice.	IC_50_: > 0.47 μM	Clinical trial terminated due to business reasons (NCT02696291).	[56,57,58]
Ivermectin	Inhibited DENV2 replication in vitro.	IC_50_: 0.5 μM	Initial unpublished findings suggest no reduction in viremia (NCT02045069).	[59,60,61]

N/A: Not available or not applicable. EC_50_: half-maximal effective concentration. IC_50_: half-maximal inhibitory concentration.

**Table 2 viruses-12-00367-t002:** Antiviral compounds targeting the E protein.

Compound	Serotype Inhibited	IC_50_ (µM)	EC_50_ (µM)	Other Flaviviruses IC_50_ or EC_50_ (µM)	Ref
Rolitetracycline	DENV2*	67.1	N/A	N/A	[109]
Doxycycline	DENV2*	55.6	N/A	N/A	[109]
P02	N/A	N/A	N/A	YFV, IC_50_: 13–17	[110]
NITD-448	DENV2*	6.8	N/A	N/A	[111]
A5	DENV2*	1.2	N/A	WNV, IC_50_: 1.6 YFV: IC_50_: 3.8	[112]
R1	DENV2*	4	N/A	N/A	[113]
Compound 1	DENV^#^	N/A	1.69	N/A	[114]
Compound 2	DENV^#^	N/A	0.90	N/A	[114]
Compound 6	DENV1-4	N/A	DENV1: 0.108 DENV2: 0.068 DENV3: 0.496 DENV4: 0.334	YFV, EC_50_: 0.47 WNV, EC_50_: 1.42 JEV, EC_50_: 0.564	[114]
3e	DENV1-4	N/A	DENV1: 0.87 DENV2: 0.85 DENV3: 0.56 DENV4: 2.5	N/A	[116]
3h	DENV1-4	N/A	DENV1: 0.58 DENV2: 0.81 DENV3: 0.39 DENV4: 0.87	N/A	[116]

* The only serotype evaluated. #The DENV serotype evaluated was not mentioned. N/A: Not available or not applicable.

**Table 3 viruses-12-00367-t003:** Antiviral peptides targeting the E protein or mimicking the action of the E protein.

Peptide	Sequence	IC_50_ (µM)	Serotype Inhibited	Ref
EF	EF	96	2	[123]
DET2	PWLKPGDLDL	500	2*	[124]
DET4	AGVKDGKLDF	35	2*	[124]
1OAN1	FWFTLIKTQAKQPARYRRFC	7	2*	[126]
DN59	MAILGDTAWDFGSLGGVFTSIGKALHQVFGAIY	2–5	1,2,3,4	[127,128]
DV2^413–447^	AILGDTAWDFGSLGGVFTSIGKALHQVFGAIYGAA	N/A	2*	[129]
P4	CKIPFEIMDLEKRHV	19	2^#^	[100]
P7	GVEPGQLKLNWFKK	13	1, 2^#^	[100]
DN57opt	RWMVWRHWFHRLRLPYNPGKNKQNQQWP	8	2*	[126]
DN81opt	RQMRAWGQDYQHGGMGYSC	36	2*	[126]

*The only serotype evaluated. #Only evaluated against DENV1 and 2. N/A: Not available or not applicable.

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
