# Peer review of "Structure-Based Design of Antivirals against Envelope Glycoprotein of Dengue Virus"

_viruses, 2020, doi:10.3390/v12040367_

Round 1

Reviewer 1 Report

The manuscript by Ishtiaq et al. entitled “Structure-based design of antivirals against envelope glycoprotein of dengue virus” provides a literature review regarding the attempts to identify antiviral strategies to combat Dengue virus infection by targeting the structural E protein of the virus. Although this review is focused on a very important medical problem, there are several weaknesses that need to be addressed:

1) The manuscript should be edited for grammatical errors, as there are many throughout the manuscript.  Verb tense is often incorrect as well as verb plurality.

2) The description of the dengue virus E protein is very lengthy and often difficult to follow.  Color pictures may help this confusion. 

3) The authors state in lines 255-256 that limitations in viremia measurements were that clinical isolates grew poorly in culture or formed inconsistent plaques during clinical trials.  However, most clinical trial measure viremia using qPCR instead assaying for infectious virus.  Moreover, others correlate the presence of NS1 to viremia.  These preferred methods would make the statement made by the authors inaccurate.

4) In lines 266-267 the authors list two approved HCV protease inhibitors as examples of antiviral success with compounds that target viral non-structural proteins. The protease inhibitors listed are odd choices to this reviewer, as these aren’t often used clinically and other examples are more relevant.  This is just a minor criticism.

5) In lines 271-272 the authors state that viral enzymes such as the NS5 protein and NS2B-NS3 may not serve as good drug targets as balapiravir and ivermectin did not reduce viremia in dengue patients. There are many problems with this statement.  First, balapiravir is a nucleoside polymerase inhibitor.  It may not have been successful in clinical trials for many reasons including sub-optimal PK, phosphorylation into the active tri-phosphate moiety may not occurred at the sites of viral replication (see ref 65 in this review), etc.  Not all NS5 inhibitors, whether nucleoside inhibitors or non-nucleoside inhibitors, are created equally.  Thus, issuing a blanket statement that NS5 is not a good target based only on the results from a single compound is not appropriate. 

                Secondly, has ivermectin been shown to target the NS3 protease in DENV?  It is a repurposed anti-parasitic agent and may likely have off target effects that are not specific to the NS3 protease.  Therefore, it is not accurate to base the clinical utility of anti-DENV protease agents on ivermectin. 

6) Why is directly engaging the E protein extracellularly instead of targeting other intracellular host proteins a main advantage of targeting the E protein? Do the authors mean to say that cellular penetration of these anti-E agents is not a concern in the drug development process? If so, this was not specifically stated.

7) Is there a reference for lines 313-314 backing up the statement that structural and functional information of DENV E protein has accelerated the screening for DENV antivirals?

8) In line 320, what is the difference between viral growth and viral replication assays? Since viruses are not alive, it is technically not correct to state that viruses grow.

9) It is unclear what the authors mean in lines 328-329, why were these compounds tested against yellow fever virus if they were identified from structure-based screening against DENV E protein?  What is a viral growth versus viral replication assay?

10) RSV is not a flavivirus as stated in line 334.

11) The units of EC50/IC50 values are missing in this 6.1 section (for example lines 335 and 363).

12) Line 337 is confusing. What does it mean to inhibit the induction of DENV-infected insect cell fusion following low pH trigger?  If the cell is already infected where does the cell fusion come in to play?

13) In lines 349-351, why again were these structure based compounds identified for DENV only assessed for activity against YFV? Also, how were these assays conducted (pcr, infectious virus yield, etc)?

14) What virus do the EC50 values reported in line 355 correspond to?

15) Is the hydrophobic pocket of the E protein highly conserved among all four DENV serotypes?

Reviewer 2 Report

Anasie et al. have provided a review of the current status of antiviral development against the clinically important dengue virus (DENV). The authors have discussed the progress of DENV antivirals currently in pre-clinical development as well as those that have reached clinical trials. However, there is a specific emphasis on the design and development of compounds that target the viral envelope glycoprotein (E protein) as this represents a suitable pharmaceutical strategy.

Despite covering a significant amount of literature, this reviewer believes that the text contains extraneous detail in particular sections. On the whole, the text is largely a regurgitation of in vitro and in vivo data from DENV antiviral studies with limited critical analysis. Rather than combining closely related studies together to form a concise opinion, the authors mostly refer to individual studies without a general critique of the literature. The authors should more clearly explain the flaws and strengths of promising antiviral candidates or particular groups/classes of compounds. In summarising the efficacy of DENV antivirals, the authors mostly provide a one-line throw away statement that offers no unique perspective but rather a shallow conclusion of the study they refer to.

Lastly, the grammar is poor on occasion throughout the manuscript and needs to be corrected. Including but not limited to minor changes to the plurality of words and the tense used in sentences.

I have provided specific examples of issues that arose whilst reviewing the manuscript:

1) Lines 28-33. There is a lack of introduction to Dengvaxia. What countries is it approved? When was it approved? What are the side effects of vaccination? Does it increase risk for other pathology inc. superinfection?

2) After 7-pages into the manuscript, the authors then begin to discuss content related to the manuscript title. The manuscript should be a lot more succinct and sections rearranged. For example, the detail on envelop protein structure is excessive and could be reduced in half. Also, section 5 and 6 should go after the introduction in section 1 as this properly leads into the grit of the text.

3) Lines 45-49. There is a lack of data. What is the evidence for the statements made? Where is the data?

4) Section 3 is overkill. There is far too much information which does not focus on the main topic of the manuscript. Despite excessive information, this section still is missing a lot of critical information which makes the manuscript difficult to follow. For example, EH1/EH2, ET1/ET2, EDI/EDII/EDIII are not explained in sufficient detail to follow the text smoothly. Hardly any information is provided.Are these acronyms? These section of glycoprotein E could be presented as a schematic to better appreciate the role of these protein domains.

5) Can the authors label the structures within Fig. 2-5 and include more information. What do the different shades represent? Where is the C-term, the N-term? What serotype are these structures from? Update any new information into the figure legend.

5) Please remove the word 'quasispecies' on line 172. This term is no longer used in the viral evolutionary field given the poor understanding and misinterpretation of this theory by the greater virology community.

6) Figure 5 would be more simple and easier to comprehend if only the monomeric structure was presented.

7) I would ask the authors to revise the legends for figures 2-6 figures as they look remarkably similar to figures present in other DENV publications. If the authors have only slightly modified others work, then they need to give credit and reference. Figure 6 in particular is almost the same as a figure in another DENV antiviral review.

8) Section 4 has no references which is unacceptable given the large amount of previous work from others in the field that has been described. Please include work of your colleagues.

9) I believe that section 5 needs expanding. This is the crux of the review. The authors should have gone into more detail regarding the in vitro and in vivo efficacy of the compounds that they listed on lines 239-240.

10)Lines 260-264 sounds like repetition of text in section 1. This is clearly why section 5 and section 6 should follow section 1. This is much more logical, rather than explaining the intricacies of E protein and then discussing antivirals that have already been explored. The text should essentially read (no antivirals/controversial vaccine --> antivirals in development --> E protein --> structure based design). The manuscript needs rearrangement so the flow is more logical.

11) The authors should also mention that hepatitis C virus antivirals approved for clinical usage also include NS5A, NS5B inhibitors. The authors state only protease inhibitors but I think they should appreciate that a multi-inhibitor approach is what has made HCV antiviral therapy so successful. The authors have also not shed light on the use of different types of antivirals in combination as a potential avenue for the treatment of DENV infections.

12) Line 271-274 is a strong claim, although the evidence presented is weak. It is not fair to claim that NS3 and NS5 inhibitors are unlikely to be useful against DENV based solely on clinical trials for two compounds. This is a naive statement. Other flaviviruses including HCV can be targeted effectively using NS3 and NS5 inhibitors, but only after incredible numbers of compounds were tested in vitro, in vivo and pushed through clinical trials. The current therapies we have today for HCV are a result of pushing thousands of compounds through in vitro and in vivo setting and dozens up dozens of compounds through clinical trials. The authors should revise this statement or exemplify their argument more.

13) Lines 298-312. This text is not necessary. It is somewhat self-explanatory from Fig. 7A and this is not a novel technique. It is widely established and does not need expansive explanation.

14) The authors have not compared the potency of the described DENV antivirals with clinically approved antivirals for other flaviviruses. This is one of the biggest issues with the manuscript and evidence of the lack of critical analysis. The authors simply state the IC50 values but don't provide any impression as to whether such potency is suitable for continued development/refinement and/pr whether compounds stand a chance of being used therapeutically. For example, one way to make that assessment to compare the potency to HCV antivirals that are now used at the bedside. Most of these compounds displayed nM to pM and sometimes fM potency with regards to in vitro and in vivo settings which allowed for very good pharmacokinetic profiling and safety given the high specificity for HCV and subsequent low dosage required for therapy. Do the authors believe the low potency of all the described DENV antivirals is promising? Which ones should be focused on? What are the similar features amongst the most promising compounds? What is the authors critique, rather than merely stating "this is what has been done".

Reviewer 3 Report

The authors provide a good review of the dengue virus envelope glycoprotein and a detailed discussion of several antiviral compounds.  Overall, the format of the manuscript flows logically and is easy to read.  I have a few very minor comments/questions that could potentially enhance the review, but even without the comments, the manuscript is very good. 

Rewrite sentence line 48-49 - just a bit more information on why the capsid inhibitor should be pursued. line 77 - the domains are not continuous. line 236 - reword the statement as follows - "The main rationale for development of a DENV antiviral is to reduce viremia during the early phase of the infection, which is expected to prevent progression to DHF/DSS. line 314 - reword the statement.  More info on what part of the E protein is "occupied" by the detergent - or at least add the word "pocket" Line 331-346- suggest to separate into two paragraphs. Can the figures be in color?

Round 2

Reviewer 1 Report

The authors have done an adequate job making the suggested changes to their manuscript.  However, there are a few minor suggestions that should be addressed relating to the changes made in the last round of edits.

1) In lines 124-125, the authors state that ribavirin displayed synergistic antiviral effects against DENV2 in vitro and in a murine model.  Synergy relates to using two compounds in combination; however, a second drug was not mentioned here.  Did the authors mean synergy or did they instead mean strong antiviral effects?

2) In lines 157-158 the authors state that other approved drugs such as velpatasvir (which is spelled incorrectly in the manuscript) and glecaprevir should be evaluated for anti-DENV activity.  Velpatasvir is an NS5A inhibitor for HCV.  DENV does not have an NS5A protein, so is there any indication that this drug is appropriate for DENV?  Additionally, are the proteases between HCV and DENV homologous, and if not, is there any other evidence that NS3/4A protease inhibitors against HCV would be effective against the NS2B/NS3 DENV protease?  The authors may wish to remove this statement.

3)  This is a minor suggestion, but in lines 180-182 the authors state that ST-610 potently inhibited NS3 helicase but exhibited a modest reduction of viremia in dengue-infected mice.  Could this be do to poor PK profiles in the mice?  Or does the drug not get into cells if the activity was only assessed in the naked protein assay? Is it worth discussing, as the authors did in the subsequent sentences, that in vitro activity may not translate to in vivo due to poor PK profiles or differences in assay methods and this may also be an example.  
